# A Simple Fabrication of Sb_2_S_3_/TiO_2_ Photo-Anode with Long Wavelength Visible Light Absorption for Efficient Photoelectrochemical Water Oxidation

**DOI:** 10.3390/nano12193444

**Published:** 2022-10-01

**Authors:** Fei Han, Sai Ma, Dong Li, Md Mofasserul Alam, Zeheng Yang

**Affiliations:** 1School of Chemistry and Chemical Engineering, Hefei University of Technology, Hefei 230009, China; 2School of Material Science and Engineering, North Minzu University, Yinchuan 750021, China; 3Key Laboratory of Polymer Materials and Manufacturing Technology, North Minzu University, Yinchuan 750021, China

**Keywords:** Sb_2_S_3_/TiO_2_, photo-anode, long wavelength, photo-electrochemical, water oxidation

## Abstract

An Sb_2_S_3_-sensitized TiO_2_ (Sb_2_S_3_/TiO_2_) photo-anode (PA) exhibiting a high photo-electrochemical (PEC) performance in water oxidation has been successfully prepared by a simple chemical bath deposition (CBD) technique. Herein, the Raman spectra and XPS spectrum of Sb_2_S_3_/TiO_2_ confirmed the formation of Sb_2_S_3_ on the TiO_2_ coatings. The Sb_2_S_3_/TiO_2_ photo-anode significantly shifted the absorption edge from 395 nm (3.10 eV) to 650 nm (1.90 eV). Furthermore, the Sb_2_S_3_/TiO_2_ photo-anode generated a photo-anodic current under visible light irradiation below 650 nm due to the photo-electrochemical action compared with the TiO_2_ photo-anode at 390 nm. The incident photon-to-current conversion efficiency (IPCE = 7.7%) at 400 nm and −0.3 V vs. Ag/AgCl was 37 times higher than that (0.21%) of the TiO_2_ photo-anodes due to the low recombination rate and acceleration of the carriers of Sb_2_S_3_/TiO_2_. Moreover, the photo-anodic current and photostability of the Sb_2_S_3_/TiO_2_ photo-anodes improved via adding the Co^2+^ ions to the electrolyte solution during photo-electrocatalysis.

## 1. Introduction

PEC water splitting using semiconductors has been expected to be one of the ideal ways to realize the conversion of solar radiative energy into hydrogen energy [1,2,3]. In the PEC system, water oxidation is considered the kinetic control step of the whole water splitting process due to the sluggish kinetics of PEC water oxidation through photo-anodes (pAs). Therefore, developing a robust, stable, and economical photo-anode is critical for highly efficient solar water-splitting.

PEC water oxidation using n-type TiO_2_ semiconductor pAs has gained considerable attention since 1972 due to their chemical and electrochemical stability in oxidation conditions, easy preparation, and noticeable incident light-to-current conversion efficiencies [4]. However, TiO_2_ pAs-based PEC water oxidation is limited because of two serious drawbacks. First, TiO_2_ has a wide bandgap (3.0–3.2 eV) that solely responds to the ultraviolet fraction of the solar spectrum (accounts for just 5% of solar irradiation) [5,6]. Second, a high recombination probability of electron-hole pairs leads to decreased incident light-to-current conversion efficiency.

Considering these disadvantages, developing highly active TiO_2_ pAs is integral to absorbing solar irradiations from ultraviolet to visible light. So far, many efforts have been devoted to noble metal loading [7,8,9,10], semiconductor recombination [11,12,13,14,15], doping [16,17,18,19] and sensitization [13,19,20,21,22,23,24,25,26,27,28,29,30] to enlarge the spectral response range due to improve its PEC performance for efficient water oxidation and to promote electron-hole pair separation. Among these, the sensitization of TiO_2_ utilizing chalcogenide-based (Sb_2_S_3_) catalysts has received significant attention [19,20,21,31,32]. Sb_2_S_3_ possesses a suitable bandgap (1.6–1.9 eV) and a high absorption coefficient (105 cm^−1^) in the visible region, which benefits from absorbing the whole visible and near-infrared range of the solar spectrum. Moreover, Sb_2_S_3_ has low toxicity and moisture, making it one of the most potential sensitizers to investigate the sensitization application. Due to these phenomena, Sb_2_S_3_/TiO_2_ pAs were fabricated by a sensitization process with Sb_2_S_3_ as a sensitizer, exhibiting enhanced PEC water oxidation properties.

Meanwhile, different strategies have been noted for preparing the Sb_2_S_3_/TiO_2_ pAs, such as atomic layer deposition (ALD), chemical bath deposition (CBD), successive ionic layer adsorption and reaction, etc. However, the costs associated with these physical methods are comprehensive. CBD has been selected in this article due to its low cost and simple procedure [33,34,35,36]. Moreover, several synthetic strategies need subsequent complex steps involving a somewhat complicated procedure for reproducible Sb_2_S_3_/TiO_2_ and developing a simple and reproducible fabrication technique for Sb_2_S_3_/TiO_2_ pAs synthesis. Herein, we describe a simple, low-cost, straightforward strategy for fabricating Sb_2_S_3_/TiO_2_ pAs using a CBD technique. Sb_2_S_3_ were coated and deposited on the TiO_2_ coated previously squeezed onto the ITO substrate utilizing a commercially available TiO_2_ paste (PST-18NR). The Sb_2_S_3_/TiO_2_ pAs show a higher PEC execution in water oxidation than TiO_2_ pAs due to the charge separation and transportation acceleration phenomena.

## 2. Materials and Methods

### 2.1. Materials

Antimony (III) chloride (SbCl_3_), Marpolose (60MP-50), HCl, and Polyethylene glycol (PEG, molecular weight = 2000) were purchased from Aladdin’s Reagent (Shanghai Aladdin Bio-Chem Technology Co., Ltd., Shanghai, China) and TiO_2_ paste (18 NR-T) was purchased from Macklin Reagent (Shanghai Macklin Biochemical Co., Ltd. Shanghai, China). An Indium tin oxide (ITO)-coated glass substrate was obtained from Dalian HeptaChroma Co., Ltd., (Dalian, China); Millipore water (DIRECT-Q 3UV, Merck Ltd. Shanghai, China) was used for all the experiments. All other chemicals were of analytical grade and were used as received unless mentioned otherwise.

### 2.2. Fabrication of TiO_2_/ITO Electrode

In a typical procedure, the TiO_2_ pAs was synthesized as follows. Before coating, the FTO-glass substrate (1.0 cm^−2^ area) was cleaned up using deionized water, acetone, and ethanol by sonication for 15 min. Then TiO_2_ paste was squeezed over an ITO-glass substrate by a doctor-blade coater and dried at 80 °C for 15 min. After repeating the procedure twice, the electrode was calcined at 550 °C under an air atmosphere for 2 h to create a TiO_2_ film-coated ITO electrode (TiO_2_/ITO).

### 2.3. Fabrication of Sb_2_S_3_/TiO_2_/ITO Electrode

SbCl_3_ (787.1 mg, 3.46 mmol) was solved in acetone (3.03 mL); after the Na_2_S_2_O_3_ (30.3 mL, 1 mol/L) solution was added, the Sb_2_S_3_ suspension was obtained. Then, the TiO_2_/ITO was immersed into the resulting Sb_2_S_3_ suspension for 24 h in the refrigerator at 10 °C. The electrode was washed thoroughly with water and air-dried to obtain the Sb_2_S_3_/TiO_2_/ITO electrode.

### 2.4. Measurement

A Shimadzu XRD-6000 diffractometer (Shimadzu International Trade (Shanghai) Co., Ltd., Shanghai, China) with Cu Kα (λ = 0.15406 nm) radiation was utilized to determine the structure of samples and to obtain the X-ray diffraction (XRD) patterns. Raman microspectroscopic apparatus (Horiba-Jobin-Yvon LabRAM HR, Paris, France) was used to obtain the Raman spectra. At room temperature, a silicon reference sample was calibrated with a theoretical position of 520.7 cm^−1^. The spectrophotometer (Shimadzu UV-2700, Shimadzu International Trade (Shanghai) Co., Ltd., Shanghai, China) was used to test the UV–visible diffuse reflectance spectra (DRS). An electron probe microanalysis (JED-2300, JEOL, Tokyo, Japan) operating at 10 kV (acceleration voltage) was used to obtain the energy-dispersive X-ray spectroscopic (EDS) data. The XPS spectra were performed using a Thermo Fisher Scientific ESCALAB Xi+ instrument (Thermo Fisher Scientific (China) Co., Ltd., Shanghai, China) and calibrated in reference to C 1 s peak fixed at 284.2 eV.

All PEC measurements were examined using an electrochemical workstation in a single-compartment PEC cell (Shanghai Chenhua Instrument Co., Ltd. Shanghai, China, CHI760E). The prepared electrode was used as the working electrode. An Ag/AgCl electrode was used as the reference electrode, and a Pt wire as the counter electrode. An aqueous 0.1 M phosphate solution (pH 6.0) was used as an electrolyte in the cell compartments and Ar-gas environments. The cyclic voltammogram (CV) was recorded at a scan rate of 50 mV s^−1^ at room temperature. The output of light intensity was calibrated as 100 mW cm^−2^ using a spectroradiometer (USR-40, Ushio Shanghai Inc., Shanghai, China). Light (λ > 500 nm, 100 mW/cm^2^) was irradiated from an irradiation source of a 500 W xenon lamp (Optical Module X, Ushio Shanghai Inc., Shanghai, China) with a UV-cut filter (λ > 500), and a 0.2 M CuSO_4_ was used as a liquid filter for cutting the heat ray. The linear sweep voltammograms (LSV) were measured at a scan rate of 5 mV s^−1^. A monochromic light with a 10 nm bandwidth was used with a 500 W xenon lamp using a monochromator for incident photon-to-current conversion efficiency (IPCE) measurements.

Photo-electrocatalysis was executed under the potentiostatic conditions at −0.3 V at 25 °C with the illumination of light (λ > 500 nm, 100 mW/cm^2^) for 1 h. 0.1 mM Co(NO_3_)_2_·6H_2_O was dissolved in the electrolyte solution to test the effect of Co^2+^ ions on photo-electrocatalysis.

## 3. Results

### 3.1. Optimization of Immersion Time

The UV–vis absorption spectrum of Sb_2_S_3_/TiO_2_/ITO electrodes immersing at different times is shown in Figure 1a. No response was observed for the TiO_2_/ITO electrode in the visible region. On the contrary, a dramatic response was observed for Sb_2_S_3_/TiO_2_/ITO electrodes, which are extended over the whole visible region (450–800 nm), demonstrating that the hole-electron pairs can be separated efficiently after sensitizing with Sb_2_S_3_. Figure 1b shows the plots of the immersion times with the absorbance at 460 nm for the Sb_2_S_3_/TiO_2_/ITO electrodes. The absorbance exhibited an increasing tendency with immersion time increased; the highest absorbance was observed when the immersion time was 24 h. After that, the absorbance decreased when the immersion time increased, suggesting that the optimum immersion condition was 24 h, which was employed in the following experiments.

### 3.2. Characterization Structure of Different Samples

The crystalline structures of the ITO, TiO_2_ and Sb_2_S_3_/TiO_2_ electrodes were characterized by XRD diffraction (Figure 2). Figure 2Ia shows the peaks of the ITO substrate. The diffracted peaks of TiO_2_/ITO (Figure 2Ib) and Sb_2_S_3_/TiO_2_/ITO electrodes (Figure 2Ic) were clearly detected at 2*θ* = 25.1°, 37.7°, 47.8°, 53.9° and 53.9°, respectively (JSPDF number: 89–4921) [5]. No peaks in terms of Sb_2_S_3_ were observed for both electrodes. However, it should be noted that two slightly broadened peaks were detected around 2*θ* = 12 and 55° for the Sb_2_S_3_/TiO_2_/FTO electrode. Raman spectroscopy was carried out for different powders scratching from the electrodes. Both TiO_2_
**(**Figure 2IIb) and Sb_2_S_3_/TiO_2_ (Figure 2IIc) samples exhibited the typical peaks located at 147, 199, 400, 517, and 638 cm^−1^, which correspond to the vibrational modes of pure anatase TiO_2_ [20,37]. Figure 2IIa shows the peak of 289 cm^−1^ for Sb_2_S_3_, which could be assigned to the Sb-S stretching modes [20,31,38,39,40]. Moreover, this broad peak was observed in Sb_2_S_3_/TiO_2_, while it was not detected for pure TiO_2_ (Figure 2IIb), confirming the Sb_2_S_3_/TiO_2_ composite formation.

The valence state of Sb_2_S_3_/TiO_2_ was investigated utilizing XPS. As shown in the survey scan in Figure 3a, Sb, Ti, S, and O elements were verified. In the high-resolution Ti 2p spectrum (Figure 3b), two peaks at 458.8 and 464.5 eV were observed, assigned to the binding energies of Ti 2p_3/2_ and Ti 2p_1/2_ of the TiO_2_ lattice, respectively [16,17,37]. The XPS spectrum of Sb_2_S_3_/TiO_2_ exhibited two peaks in the S 2p region at 161.4 and 162.4 eV correspond to S 2p_1/2_ and S 2p_3/2_ of divalent S in Figure 3c [16,19,41]. The peaks at 529.3 and 538.7 eV in the Sb 3d region for Sb_2_S_3_/TiO_2_, respectively, are assigned the Sb^3+^ oxidation state of Sb and are consistent with the earlier-reported values of Sb_2_S_3_ [19,42,43,44]. The relatively weak peak at 531.7 can be assigned to the formation of the Sb-O band at the surface, possibly associated with the oxygen atom of TiO_2_.

The FE-SEM observed the morphologies of the TiO_2_ and Sb_2_S_3_/TiO_2_ electrodes. Figure 4I(a) exhibited the top view of the TiO_2_ electrode at low magnification, in which a continuous uniform and smoothed surface were formed. This would be beneficial to improve the PEC performance of the Sb_2_S_3_/TiO_2_ electrode due to accelerating the water oxidation reaction at the surface of the electrolyte solution. The FE-SEM images at high magnification (Figure 4I(b)) showed that the average particle size of TiO_2_ was 10 nm. The surface FE-SEM image of the Sb_2_S_3_/TiO_2_ electrode at low magnification is shown in Figure 4I(c), where the larger Sb_2_S_3_ particles were deposited on the TiO_2_ surface. After that, the 60–100 nm Sb_2_S_3_ particles were clearly observed in Figure 4I(d). EDX analyses of the Sb_2_S_3_/TiO_2_ electrode were taken to confirm the Sb_2_S_3_ deposition and complete the chemical composition information. The elemental maps of the EDX for the Sb_2_S_3_/TiO_2_ electrode are shown in Figure 4II. The Ti and O mapping signals (Figure 4II(c,d)) and S and Sb (Figure 4II(e,f)) were detected. The distribution of Sb_2_S_3_ onto the TiO_2_ in the Sb_2_S_3_/TiO_2_ electrode appeared clearly.

### 3.3. The Optical Properties of Sb_2_S_3_/TiO_2_

The diffuse reflection spectrum of TiO_2_ and Sb_2_S_3_/TiO_2_ are presented in Figure 5a. It was observed that the Sb_2_S_3_/TiO_2_ absorb the visible light below 650 nm, exhibiting significant redshift in a broad wavelength region compared to TiO_2_ (395 nm). Tauc plots based on UV–vis DRS are exhibited in Figure 5b. The TiO_2_ showed a single absorption edge of 3.20 eV, corresponding to the previous report relating to anatase TiO_2_ [5]. The Sb_2_S_3_/TiO_2_ provided two different slops due to the appearance of the new absorption edges, one of the band energies was 1.94 eV, and another was 2.37 eV. The photo-response of the Sb_2_S_3_/TiO_2_ was significantly improved in the visible region. This is attributed to the existence of Sb_2_S_3_, which significantly promotes the separation of the electron-hole pairs.

### 3.4. Photoelectrocatalytic Properties

The action spectra of IPCE were recorded at −0.3 V by utilizing TiO_2_ and Sb_2_S_3_/TiO_2_ electrodes, as shown in Figure 5c. The IPCE value at 400 nm for Sb_2_S_3_/TiO_2_ electrode (7.7%) is 37 times higher than that for TiO_2_ electrode (0.21%). Further, the onset wavelength for photo-current generation on Sb_2_S_3_/TiO_2_ electrode was 650 nm (1.9 eV), which is noticeably longer than the TiO_2_ electrode (400 nm, 3.1 eV). The IPCE action spectra consisted of the UV-DRS data for Sb_2_S_3_/TiO_2_ electrode, indicating that the pC was generated based on the bandgap excitation through collateral excitation from Sb 3d orbital to CB.

Several PEC examinations were performed to reveal the intrinsic PEC performances of TiO_2_ and Sb_2_S_3_/TiO_2_ electrodes towards water oxidation, as shown in Figure 6. Figure 6a showed the CV of the TiO_2_ (black) and Sb_2_S_3_/TiO_2_ (red) electrodes in a 0.1M phosphate solution (pH = 6.0). Both electrodes did not have photo-anodic currents in the dark condition (dashed line). Contrary to the TiO_2_ electrode, which showed a 9.1 A/cm^2^ at 0.2 V, the Sb_2_S_3_/TiO_2_ electrode generated a stable photo-anodic current of 42.3 A/cm^2^ based on water oxidation over 0.6 V, showing 4.6 times higher value than that of the TiO_2_ electrode. Such remarkable enhancement of pC density for the Sb_2_S_3_/TiO_2_ electrode can be ascribed to the improvement of light absorption in the visible region. The LSV of two electrodes shown in Figure 6b was taken under chopped/simulated chopped visible light irradiation. Interestingly, it can be observed that the photo-responses of anodic and cathodic currents for the Sb_2_S_3_/TiO_2_ electrode (red) were clearly observed above or below −0.5 V, although hardly any pAs current was observed for the TiO_2_ electrode (black, insert).

The results suggest that the photo-current direction of the Sb_2_S_3_/TiO_2_ electrode can be modified according to the applied potential. When the potential is below −0.5 V, the charges separation based on bandgap excitation has caused upon irradiation with visible light, then the electrons from ITO are rapidly injected into the valence band of Sb_2_S_3_ through the conduction band of TiO_2_. Afterward, the electron acceptor in the conduction band of Sb_2_S_3_ is reduced by the electrons to generate a cathodic current. On the other condition of potential above −0.5 V, Sb_2_S_3_ absorbs photons to generate electron-hole pairs, the electrons are injected into the electron transport layer (TiO_2_) and collected by ITO, and the holes oxidize the water at the surface of Sb_2_S_3_ to generate an anodic current. As the light was turned on and off, instantaneous positive and negative transient photocurrents were observed under chopper illumination at −0.3 V (Figure 6c).

A negligible photo-current for the TiO_2_ electrode (black) was observed. However, the photo-current of 30 A/cm^2^ for the Sb_2_S_3_/TiO_2_ electrode (red) was observed. It is indicated that the charge carriers are very efficiently separated and limited to recombine after Sb_2_S_3_ sensitization. The photo-electrocatalysis was conducted under visible light irradiation (*λ* > 500 nm, 100 mW/cm^2^) at −0.3 V in a 0.1M phosphate solution (pH = 6.0) for 1 h (Figure 6d). The I-t profile of the Sb_2_S_3_/TiO_2_ electrode exhibited a photo-current of 30 A/cm^2^ over the period, while for the TiO_2_ electrode, almost no photo-current was observed. The PEC performance of hematite and WO_3_ photo-anodes for water oxidation can be enhanced by adding Co^2+^ ions into the electrolyte solution to act as catalysts at the electrode interface with the electrolyte solution without any deposition [45,46].

The influence of adding Co^2+^ ions on the PEC water oxidation for the Sb_2_S_3_/TiO_2_ electrode was measured to prove this assumption. The performance of the Sb_2_S_3_/TiO_2_ electrode for water oxidation was improved by the presence of 0.1 mm Co^2+^ ions in the phosphate buffer solution. It should be noted that the stability of the photo-current for the Sb_2_S_3_/TiO_2_ electrode by the presence of Co^2+^ ions was remarkably improved, which consequently generated a 2.7 times higher photo-current of 80 A/cm^2^ compared to the absence one. Such a lower decay rate (20%) of photo-current for the Sb_2_S_3_/TiO_2_ electrode by the addition of Co^2+^ ions demonstrates that the stability is ascribed to the acceleration of the water oxidation reaction between the Sb_2_S_3_/TiO_2_ surface and the electrolyte solution interface, which significantly reduce the recombination rate of photo-generated carriers.

## 4. Results and Discussion

Two main disadvantages are associated with high recombination of electron-hole pairs and a wide bandgap for TiO_2_. Many efforts have been devoted to noble metal loading, semiconductor recombination, doping and sensitization to enlarge the spectral response range due to improving its PEC performance for efficient water oxidation and promoting electron-hole pairs separation. Sb_2_S_3_ possesses a suitable bandgap and a high absorption coefficient in the visible region, which benefits from absorbing the whole visible and near-infrared range of the solar spectrum. Thus, the Sb_2_S_3_/TiO_2_ were fabricated by a sensitization process with Sb_2_S_3_ as a sensitizer, exhibiting enhanced PEC water oxidation properties. Herein, a simple, low-cost and straightforward strategy for fabricating Sb_2_S_3_/TiO_2_ using a CBD technique was reported in this paper.

## 5. Conclusions

An Sb_2_S_3_/TiO_2_ photo-anode was fabricated utilizing a simple and low-cost CBD strategy. XPS and elemental mapping detected Sb and S signals and Ti and O, demonstrating the formation of Sb_2_S_3_ and TiO_2_, respectively. The Sb_2_S_3_/TiO_2_ photo-anode can utilize visible light below 650 nm for PEC water oxidation, in contrast to utilization below 395 nm for the TiO_2_ photo-anode. The IPCE of 7.7% at 400 nm for Sb_2_S_3_/TiO_2_ photo-anode was higher than that of 0.21% for TiO_2_ photo-anode by 37 times. The Sb_2_S_3_/TiO_2_ photo-anode can generate a photo-anodic current and a photo-cathodic current in contrast to generating a photo-anodic current for the TiO_2_ photo-anode. The photo-electrocatalytic performance of Sb_2_S_3_/TiO_2_ photo-anode would be further improved by adding Co^2+^ ions in the electrolyte solution due to low recombination rates of charge carriers and facilitated electron transfers.

## Figures and Tables

**Figure 1 nanomaterials-12-03444-f001:**
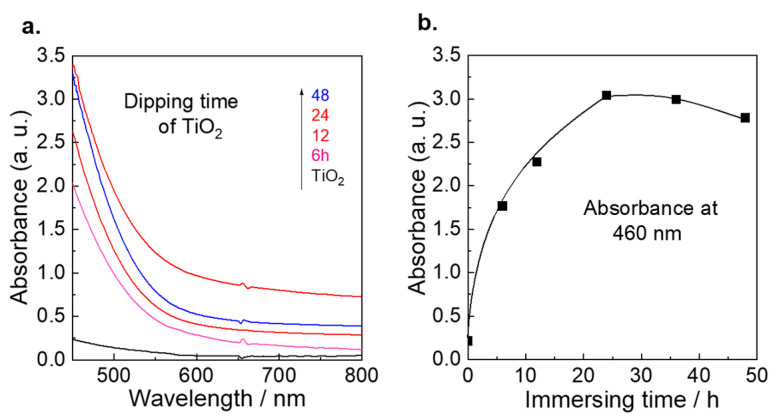
(**a**) UV–Vis spectral change of the Sb_2_S_3_/TiO_2_ at difference time for dipping; (**b**) Plots of Absorbance vs. immersion time for Sb_2_S_3_/TiO_2_, respectively.

**Figure 2 nanomaterials-12-03444-f002:**
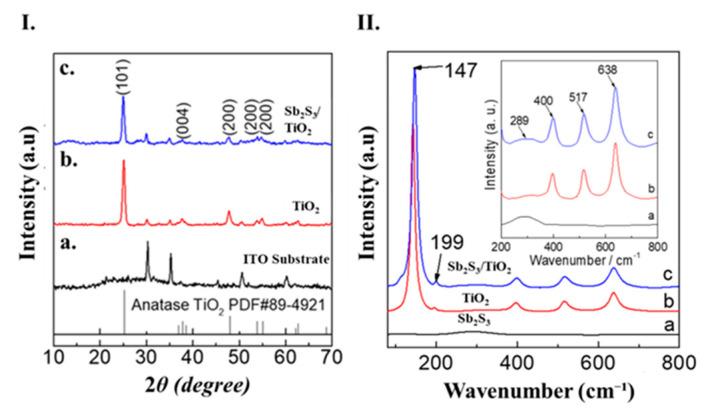
(**I**) X-RD data; (a) ITO substrate; (b) TiO_2_, and (c) Sb_2_S_3_/TiO_2_ photo-anodes, respectively; (**II**) Raman spectroscopy; (a) Sb_2_S_3_, (b) TiO_2_, and (c) Sb_2_S_3_/TiO_2_ powders, respectively.

**Figure 3 nanomaterials-12-03444-f003:**
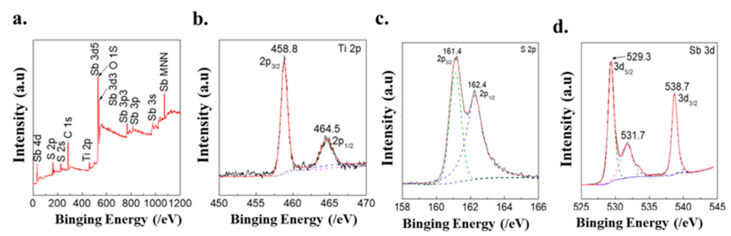
XPS spectra of the Sb_2_S_3_/TiO_2_; (**a**) survey scan, (**b**) Ti 2p, (**c**) S 2p, and (**d**) Sb 3d, respectively. For the spectra in the Ti 2p, S 2p, and Sb 3d regions, the dashed lines are the deconvoluted spectrum bands, and the solid lines are the spectrum simulated by the deconvoluted bands.

**Figure 4 nanomaterials-12-03444-f004:**
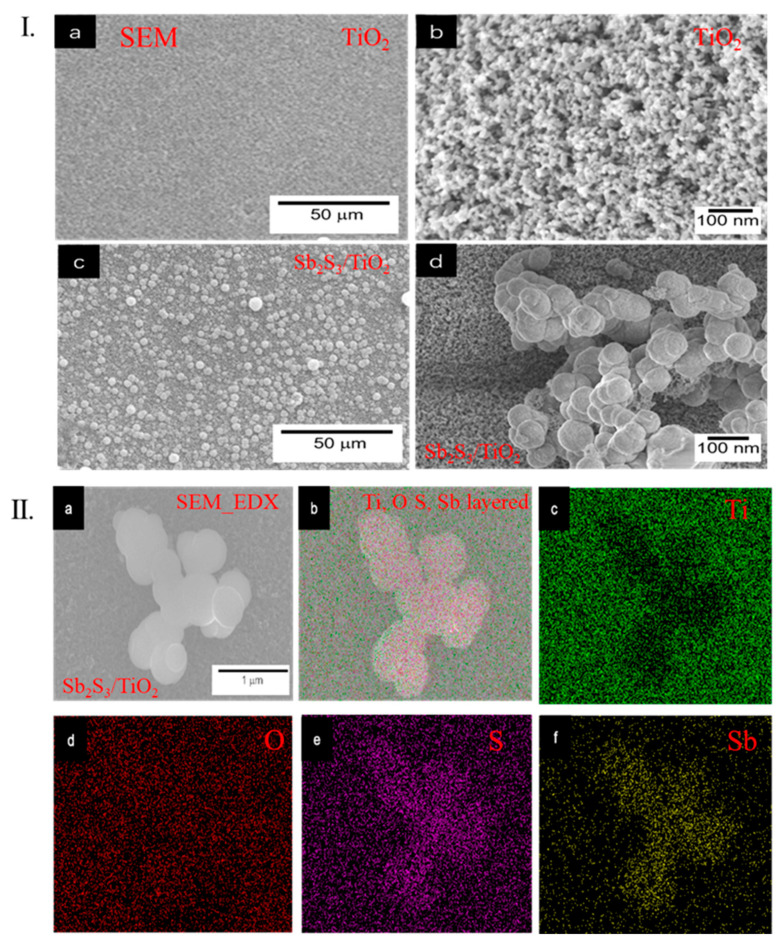
(**I**) SEM images; (**a**,**b**) TiO_2_ and (**c**,**d**) Sb_2_S_3_/TiO_2_ electrode, respectively; (**II**) (**a**) SEM_EDX elements distribution mapping images of the Sb_2_S_3_/TiO_2_ electrode; (**b**) Ti, O S, Sb layered, (**c**) Ti, (**d**) O, (**e**) S, and (**f**) Sb, respectively.

**Figure 5 nanomaterials-12-03444-f005:**
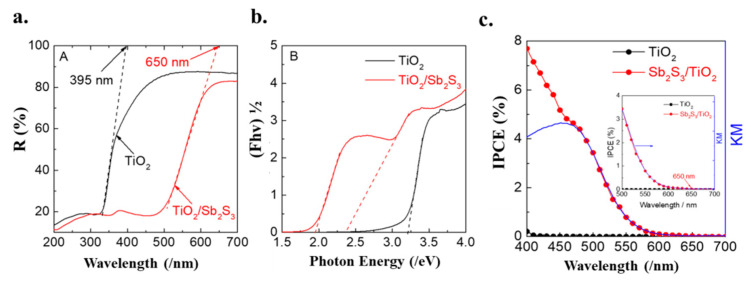
(**a**) Diffuse reflection spectrum of TiO_2_ (black) and Sb_2_S_3_/TiO_2_ (Red); (**b**) Tauc plots of TiO_2_ (black) and Sb_2_S_3_/TiO_2_ (red). The dashed lines show tangent lines near the edges of the plots, respectively, (**c**) Action spectra of IPCE of theTiO_2_ (black) and Sb_2_S_3_/TiO_2_ (red) electrode in a 0.1 M phosphate buffer solution of pH 6.0 at −0.3 V vs. Ag/AgCl. The insets show the magnified spectra near the edges.

**Figure 6 nanomaterials-12-03444-f006:**
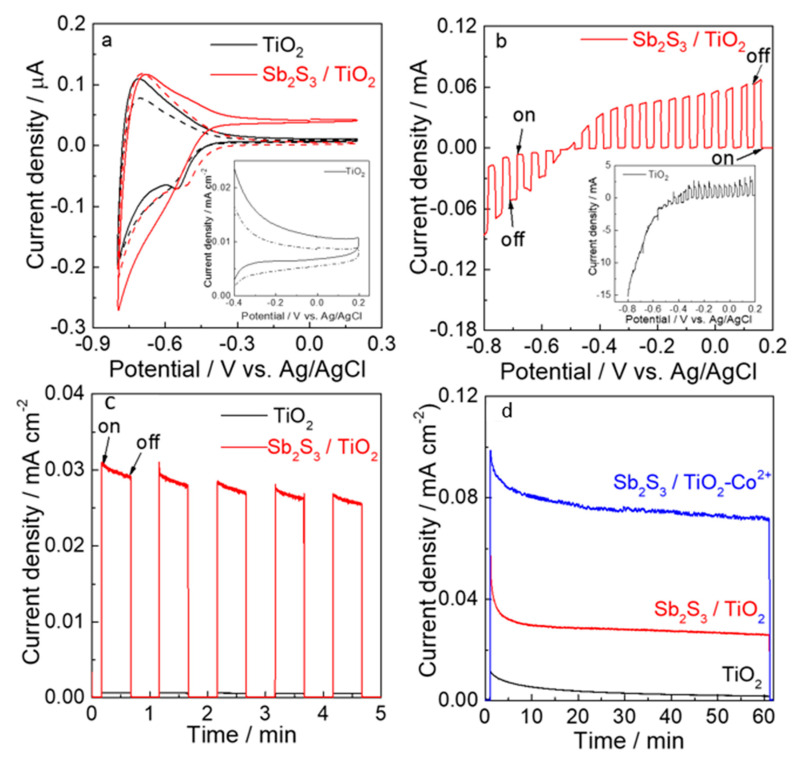
(**a**) CVs measured in the dark (dash) and on irradiation (solid); the subfigures in Figure 6a is the magnification from -0.4 V to 0.2 V for TiO_2_.(**b**) LSV plots and; (**c**) Transient on/off chopped photo-current response of the Sb_2_S_3_/TiO_2_ (red) and TiO_2_ (the subfigures in Figure 6b) photo-anode in a 0.1 M phosphate buffer solution of pH 7.0 with visible light irradiation (λ > 500 nm, 100 mW/cm^2^) chopped. (**d**) photo-current density-time profiles for Sb_2_S_3_/TiO_2_ (red) and TiO_2_ photo-anode (black) in 0.1 m phosphate buffer solution of pH 6.0 during photo-electrocatalytic water oxidation at −0.3 V vs. Ag/AgCl. Sb_2_S_3_/TiO_2_ (blue) in the presence of 0.1 mm Co^2+^ ions in the electrolyte solution.

## Data Availability

Not applicable.

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
