# Peer review of "A Simple Fabrication of Sb2S3/TiO2 Photo-Anode with Long Wavelength Visible Light Absorption for Efficient Photoelectrochemical Water Oxidation"

_nanomaterials, 2022, doi:10.3390/nano12193444_

Round 1
Reviewer 1 Report
The manuscript entitled “A Facile Fabrication of Sb2S3/TiO2 photo-anode with long wavelength visible light absorption for efficient photoelectrochemical water oxidation” by F. Han et al. reports the synthesis route of a photoanode for photoelectrochemical water splitting. I would recommend the following corrections.
1) The sentence “Moreover, Sb2S3 has low toxicity, moisture, and air stability,..” suggests that Sb2S3 has low stability in air and humid air. I do not think the authors wanted to say that.
2) What was the temperature of the suspension while it was kept in the refrigerator?
3) In Fig. 1b, there is an artificial maximum of absorbance around the 35 h time. It is, probably, a result of using a B-spline type of line to connect the points. By the way, I believe immersing time is measured in h, not in h-1.
4) Why do the authors consider “a slightly broadened peak was detected around 2θ = 55° for the Sb2S3/TiO2/FTO electrode” as a possible indication of the Sb2S3 presence? What is an approximate amount of Sb2S3? Have you estimated it e.g. by weighting the difference in mass before and after immersing followed by drying? Have you checked the XRD pattern of Sb2S3 obtained by you? XRD is not very sensitive to nano- materials even if they are crystalline.
5) Line 156 and others: It should be valence state, valence band not “valance state” etc.
6) Line 263: The first sentence does not have any verb.
7) The subchapter “Discussion” seems unnecessary. This short passage better fits into the Introduction. The discussion of the results was actually in the “Results” which could be named “Results and Discussion”.
Author Response
Response to Reviewer 1 Comments
Thank you very much for your kind and helpful comments. We revised the manuscript carefully considering your comments.
Point 1: The sentence “Moreover, Sb2S3 has low toxicity, moisture, and air stability,..” suggests that Sb2S3 has low stability in air and humid air. I do not think the authors wanted to say that.
Response 1: As you pointed out, it has been mentioned correctly in the revised manuscript.
Point 2: What was the temperature of the suspension while it was kept in the refrigerator?
Response 2: The suspension was kept at 10 ºC in the refrigerator. it has been mentioned correctly in the revised manuscript.
Point 3: In Fig. 1b, there is an artificial maximum of absorbance around the 35 h time. It is, probably, a result of using a B-spline type of line to connect the points. By the way, I believe immersing time is measured in h, not in h-1.
Response 3: In order to check the 24 h time is the the optimum immersing condition. We have added a new experiment of the 36 h time, and the figure 1b was replotted. All the changes have been modified in the revised manuscript.
Figure 1. b. Plots of Absorbance Vs. Immersing time for Sb2S3/TiO2, respectively.
Point 4: Why do the authors consider “a slightly broadened peak was detected around 2θ = 55° for the Sb2S3/TiO2/FTO electrode” as a possible indication of the Sb2S3 presence? What is an approximate amount of Sb2S3? Have you estimated it e.g. by weighting the difference in mass before and after immersing followed by drying? Have you checked the XRD pattern of Sb2S3 obtained by you? XRD is not very sensitive to nano- materials even if they are crystalline.
Response 4: The peak at 2θ = 55° was assigned to the Sb2S3/TiO2/FTO electrode according to the earlier reported literatures (J. Photoch. Photobio. A, 2015, 303-304, 8-16, Int. J. Hydrog. Energ., 2017. 42, 8418-8449, J. Mater. Sci-Mater. El., 2019. 30, 5631-5639.).
We also reaserched the weight change before and after immersing followed by drying. Compare to the weight of TiO2/ITO electrode of 1.3079 g, the increasing of weight was 0.03 g for the Sb2S3/TiO2/FTO electrode (1.6079 g).
As you pointed, The broadened peak at 2θ = 55° was observed due to the deposition of the large Sb2S3 particles of 60-100 nm, which was clearly observed in the SEM image about 60-100 nm. So, we have a reason to believe that the Sb2S3/TiO2 is nano- material.
Point 5: Line 156 and others: It should be valence state, valence band not “valance state” etc.
Response 5: It has been mentioned correctly in the revised manuscript.
Point 6: Line 263: The first sentence does not have any verb.
Response 6: It has been mentioned correctly in the revised manuscript.
Point 7: The subchapter “Discussion” seems unnecessary. This short passage better fits into the Introduction. The discussion of the results was actually in the “Results” which could be named “Results and Discussion”.
Response 7: It has been mentioned correctly in the revised manuscript.

Reviewer 2 Report
1. Why was a calcination temperature of 550 °C chosen? Why not to apply it to the antimony sulfide electrode?
2. Why a >500 nm cut filter was used? Usually, the UV component is cut by a 400 nm filter.
3. Please, present the absorption spectrum in Fig. 1(left) with wavelengths on the horizontal axis to allow facile comparison with other data.
4. At which binding energy is the principal O1s located for the pure TiO2? This because it is notoriously difficult to discern between Sb3d and O1s when both elements are present at the surface (confirmed by SEM_EDX). The peak at 531.7 eV could be due to hydroxylation of the surface.
5. Were all the curves in Fig.6 measured in phosphate buffer (please give the composition)? Also, the pH of the buffer is mentioned as 7.0 in the text and 6.0 in the figure caption, please, clarify.
Author Response
Response to Reviewer 2 Comments
Thank you very much for your kind and helpful comments. We revised the manuscript carefully considering your comments.
Point 1: Why was a calcination temperature of 550 °C chosen? Why not to apply it to the antimony sulfide electrode?
Response 1: In oder to prepare the Sb2S3/TiO2 photoanode with high efficient photoelectrochemical for water oxidation, The key is to prepare a TiO2 with high crystallinity, which wiil be perepared at 550 °C. Sb2O3 instea of Sb2S3 may be generated, once the antimony sulfide electrode is calcined, which have already reported previously (Mater. Design, 2017, 121, 1–10; Trans. Nonferrous Met. Soc. China, 2020, 30, 1625−1634.).
Point 2: Why a >500 nm cut filter was used? Usually, the UV component is cut by a 400 nm filter.
Response 2: Our principal issue is to prepare a Sb2S3/TiO2 photoanode with long wavelength visible light absorption. Of course, 400 nm cut filter can aslo be used, but we want to utilize longer wavelength visible light. In addition, the diffuse reflection spectrum of Sb2S3/TiO2 suggests that the Sb2S3/TiO2 can absorb the visible light below 650 nm, which are presented in Figure 5a. Based on these reasons, we chose to use a >500 nm cut filter.
Point 3: Please, present the absorption spectrum in Fig. 1(left) with wavelengths on the horizontal axis to allow facile comparison with other data.
Response 3: As you pointed out, it has been mentioned correctly in the revised manuscript.
Figure 1. UV-Vis spectral change of the Sb2S3/TiO2 at difference time for dipping; b. Plots of Absorbance Vs. Immersing time for Sb2S3/TiO2, respectively.
Point 4: At which binding energy is the principal O1s located for the pure TiO2? This because it is notoriously difficult to discern between Sb3d and O1s when both elements are present at the surface (confirmed by SEM_EDX). The peak at 531.7 eV could be due to hydroxylation of the surface.
Response 4: As you pointed, the apparent peaks at 531.6 eV and 530.2 eV in the XPS spectrum of O 1s can be assigned to the hydroxylation of the surface and Ti-O species, respectively. In the Sb3d region, The peak at 531.7 eV could be the formation of the Sb-O band at the surface according to the earlier reported (Int. J. Hydrog. Energ., 2021, 46, 31216-31227.)
Figure 2. XPS spectra of the O 2p of TiO2
Point 5: Were all the curves in Fig.6 measured in phosphate buffer (please give the composition)? Also, the pH of the buffer is mentioned as 7.0 in the text and 6.0 in the figure caption, please, clarify.
Response 5: As you pointed out, it has been mentioned correctly in the revised manuscript.

Round 2
Reviewer 2 Report
The authors have responded adequately to queries and the manuscript is now publishable in the present form